# The Role and Relevance of Hearing Dogs from the Owner's Perspective: An Explorative Study among Adults with Hearing Loss

Audrey Lalancette [1,2,*] , Marie-Alycia Tremblay [2] and Mathieu Hotton [1,2]

1  Centre Interdisciplinaire de Recherche en Réadaptation et Intégration Sociale (Cirris),
   Quebec, QC G1M 2S8, Canada
2  Rehabilitation Department, Université Laval, Quebec, QC G1V 0A6, Canada
*  Correspondence: aulal15@ulaval.ca

**Abstract:** This study aimed to explore perceptions and experiences about how owning a hearing dog can influence the functioning and the autonomy of people with hearing loss. Three adults participated in a semi-structured interview. The interviews were video recorded, transcribed, and coded. A procedure combining qualitative content analysis and interpretative phenomenological analysis was used. The study shows how specific aspects of hearing dogs are associated with increased autonomy and sense of security among owners. The attentive dog-owner pairing, the outstanding training and the companion role of the hearing dog are the main elements supporting the high satisfaction related by all the participants. In regard of the location context (Quebec, Canada), ongoing challenges for owners are reflected in the lack of visibility of this rehabilitation means and its poor recognition from the society, resulting in the constant burden to explain the dog's work to others. For adults with hearing loss, the hearing dog is a relevant way of offering both the benefits of functional assistance and the psychosocial support of a pet. The association between owning a hearing dog and improved overall well-being suggests that this form of rehabilitation should be considered as a pertinent option by hearing health professionals.

**Keywords:** hearing loss; hearing dog; assistance dog; service dog; rehabilitation; auditory rehabilitation; audiology; deafness

## 1. Introduction

The consequences of auditory impairment are numerous and important. It can lead to communication and speech-understanding issues, but also to weariness, anxiety, social isolation, psychological distress, and depression [1–3]. Furthermore, loved ones often feel frustrated because of the communication difficulties associated with hearing loss. They can develop an increased sense of burden, associated with the supporting role they feel they must assume [4–6]. To improve the functioning and the autonomy of people with hearing loss, hearing health professionals can recommend hearing assistance devices. The most common is the hearing aid, a technological aid known to be effective in improving communication and reducing the biopsychosocial impacts of hearing loss [2,7]. Other hearing assistance technology, i.e., devices used as an alternative or in addition to hearing aids in real-life situations where the effectiveness of hearing aids is limited, can also be recommended [8]. For example, a person with a hearing impairment may use an environmental alerting system (EAS) in conjunction with hearing aids to help detect sound alarms in their home (e.g., smoke detector, doorbell, or telephone). Because of its primary alerting function, the hearing dog is mainly seen as a complement to hearing aids or ear implants. In most cases, the choice to have a hearing dog is seen as an alternative or a complementary mean of meeting the needs of the individual when hearing aids are not

suitable for economic, socio-cultural, or psychological reasons such as the refusal of surgery (implant) or the rejection of devices [9,10].

Person-centered rehabilitation should not only focus on reducing activity limitations and participation restrictions, but also take into account the emotional and psychological needs of the individual [9]. Rehabilitation options that take these aspects into account are more likely to increase the quality of life (QoL) of the user [11]. One of these rehabilitation options is the service dog, a general term referring to dogs specifically trained to assist individuals with disabilities. Its primary purpose is to reduce activity limitations across a range of areas, including health, mobility, mood, and social interactions [12]. In the case of the hearing dog, it is specifically trained to signal to its owner the presence of environmental sounds such as fire alarms, baby cries or ringtones, both at home and outside [11–13]. When a signal is detected, the dog sits in front of its owner and brings him towards the source of the noise.

The benefits of service dogs can have far-reaching implications for individuals with disabilities, for society and for the economy by promoting independence, learning, and employability [14]. Most individuals in the process of receiving a hearing dog are regular users of hearing assistance technology such as hearing aids or EAS [9]. Higher limitations than the general population in terms of physical and general health, social functioning and emotional functioning are also reported in those individuals. In terms of expectations while on the waiting list for a hearing dog, respondents said they wanted their dog to be a companion and to be able to detect at least three sounds from their environment [9]. The benefits of the hearing dog reported in the literature seem to respond to the significant functional and social limitations reported by those users in waiting [9]. To this end, the association between the ownership of a hearing dog and an increase in QoL [14] supports the relevance for health professionals to consider these dogs as a relevant option in hearing loss rehabilitation. Various benefits have been studied in terms of psychological functioning, social functioning, health and safety, and participation [15]. However, owning a service dog also comes with responsibilities. The dog has its own personality and may not be perfect, just as its training may not be flawless [15–17].

Significant positive effects of the hearing dog in different components of functioning have been found. It has been repeatedly shown to perform well for the tasks it is trained to do, consisting primarily in alerting [16,18,19]. Interrogated about the reasons for acquiring a hearing dog, most of the owners mentioned the desire to feel more secure, after the need of being alerted by sounds that are inaudible to them [17]. This desire seems well founded: 93% of respondents in a study by Valentine et al. [20] claimed to feel more secure since acquiring their dog. In a longitudinal study, Lundqvist et al. [21,22] also identified that hearing dogs gave participants the opportunity to become more active and independent of those around them, in addition to making them feel more secure. Furthermore, a considerable number of studies focused on demonstrating the psychological, social, and emotional benefits of adopting a hearing dog, which appear to be maintained over time [12–15,17,19,20,23]. Global improvement of functioning and significant reduction in loneliness, anxiety, stress, tension, and depression have been demonstrated [15,19,20]. Hearing dogs were also found to positively affect mental well-being and to reduce social isolation and dependency on others [13]. In studies using questionnaires, it was shown that individuals with a hearing dog presented an increased QoL compared to individuals waiting to receive a hearing dog. More specifically, they demonstrated higher satisfaction on items concerning autonomy, learning, work, and physical health [14]. Their confidence, independence, and self-esteem were also increased, leading to better self-realization [14].

In contrast to these findings, Singh et al. [11] found no significant beneficial effects of owning a hearing dog. According to their results, it is possible but uncertain that the owning of such a dog brings a reduction in activity limitations or any emotional and social benefits to the QoL of individuals [11]. In fact, the results reveal a significantly lower QoL compared to individuals waiting for a dog in the questionnaires. However, several positive trends were identified in the qualitative analysis of the participants' responses. These

diverging results demonstrate that the existing research on the impacts of the hearing dog in the lives of individuals is limited and that current data do not allow robust conclusions to be drawn on the subject [13].

Improved social integration, in which the hearing dog acts as a catalyst for social interactions, have been discussed: owners reported being less inclined to avoid interactions with other individuals after the placement of the dog [19]. Thus, the presence of the dog increases the opportunities for social exchanges and meeting new people, which leads to a feeling of being accepted by society [19]. Several others have also noted an improvement in social functioning. McNicholas & Collis [23] noted that 92% of their respondents reported an improvement in their feeling of social integration, in relation to increased communication opportunities. Adding to this list of social benefits, hearing dogs were found to increase adaptation in social situations, to improve interactions with the hearing community, to strengthen social relationships and to increase social recognition [15,17,20,21]. On the other hand, others have demonstrated that individuals with a hearing dog specifically did not demonstrate significantly higher satisfaction on the items of social activities and self-understanding [14]. They suggest that reduced stigmatization and social exclusion among individuals with non-apparent physical disabilities could explain these results [14]. Another explanation given for the lack of significant improvement in social relations relates to the limited number of hearing dogs recruited in the various studies [11,14].

Although most interrogated owners reported being satisfied with the acquisition of their hearing dog, some negative aspects were also brought up. Among those are the dog's behavior problems, the lack of training, the lack of post-pairing follow-up, the additional training necessary for certain tasks specific to the owner, the making of travel arrangements, the access to public places, the responsibilities associated with maintenance and care of the dog, the distraction of the dog by people, and the failure to alert to sound [15–17]. There is also a predominant misunderstanding in society of the role of assistance dogs [21,22].

In the province of Quebec, Canada, the *Charter of human rights and freedoms* protects any person with a disability who uses a guide or assistance dog. The hearing dog being recognized by this authority since 2007, any person accompanied by such a dog has the right to access without discrimination public places, public transport and taxis, workplaces and places of leisure [24]. Despite this organizational recognition, the service dog is not considered a legitimate means of alleviating activity limitations and participation restrictions caused by hearing impairment by the authorities of the Quebec Ministry of Health and Social Services (MSSS) [25]. No specific law exists to circumscribe its use and case law is often cited to settle any disputes that may arise. *Mira Foundation*, the only recognized guide and service dogs training center in Quebec, does not offer a dog training or support program for people with hearing impairment. In addition, the *Raymond-Dewar Institute*, the main rehabilitation center specializing in hearing impairment in Quebec, confirmed "*that it is understood that the service dog for the deaf or hard of hearing is not assigned as part of a recognized rehabilitation program addressing the needs of people with a hearing impairment*" [25]. Associated with the lack of promotion and accessibility of hearing dogs, the recommendation of this assistive help as a means of auditory rehabilitation remains scarce in Quebec.

The existing literature on hearing dogs, which is very limited, comes mainly from Europe and the United States. To our knowledge, no study regarding patients' experience with hearing dogs in Canada has been published to date. The benefits and disadvantages of a partnership with a hearing dog in Canada have not been documented precisely, which is even more true for the province of Quebec that does not have local access to this service. Quebecers who want to benefit from a hearing dog need to contact the Lions Foundation of Canada Dog Guides to submit their application. Once they've been accepted, they are required to travel to one of their physical locations to complete the training process. Oakville (located in the province of Ontario, Canada) is where the *Foundation* operates its closest facility. The duration of the stay and the training for the new recipient of a dog is two to three weeks. All expenses (housing, food, travel, etc.) are covered by the Foundation. Beneficiaries must be in class all day, seven days a week so that the owner-dog team can

learn to work together. This is a rather time and energy consuming process, even more for a French Quebecer who must take a leave of work and travel to Ontario, where the English language is predominant.

The main aim of this research is therefore to shed light on a little-known means of hearing rehabilitation by providing a better understanding of the hearing dogs' owners experience. We directed our attention to the benefits and disadvantages of owning a hearing dog. Data was also collected on participants' satisfaction, social participation, and sense of security. The collected information provides a better understanding of the role and relevance of the hearing dog and assesses the extent to which it meets the needs of people with hearing loss, in complement to hearing aids. We then discussed the relevance of the hearing dog in comparison with hearing assistance technologies such as the EAS. Finally, we explored the potential relevance of setting up a distribution center for hearing dogs in Quebec province, as it currently exists in other Canadian provinces.

## 2. Materials and Methods

### 2.1. Conceptual Framework

We guided our work in consideration of the International Classification of Functioning, Disability and Health (ICF) presented by the World Health Organization (WHO) [26]. This model focuses on describing functioning, disability, and health from a multidimensional perspective by highlighting the interaction between different components. The ICF is recognized in rehabilitation research as a tool to guide the collection of data on activities, social participation, and personal and environmental barriers or facilitators. This model is used with various individuals with disabilities and has been proven a relevant research tool since it helps to describe the functioning of people with disabilities by considering the environmental and personal factors that influence it. It is also used to guide the planning and evaluation of health and rehabilitation services. On this matter, Sachs-Ericsson et al. [12] used the ICF to conduct a literature review of the benefits of assistance dogs and hearing dogs as a rehabilitation mean. They specifically looked at the effects on body functioning, activity, participation in society, and contextual factors such as the environment and personal factors, and thus demonstrated the positives effects of owning a hearing dog. The use of the ICF model in this project proves itself relevant since it makes it possible to assess the impacts of the hearing dog using a multidimensional approach to health and rehabilitation in people with hearing loss.

### 2.2. Study Design and Procedures

Previous work has demonstrated the relevance of qualitative avenue of research when studying a topic such as hearing dogs [21,22]. Thus, a cross-sectional qualitative case study design was used [27,28]. Semi-structured individual interviews were conducted with adults and seniors living in Quebec. To ensure that all relevant elements were addressed during the interviews, the health domains and categories represented in the comprehensive ICF core set for hearing loss [29] were used as a guide to develop the interview grid used during data collection. Lasting from 45 min to two hours, the interviews took place on a web conference platform. This project was approved by the ethics board of the CIUSSS Capitale-Nationale (project #2022-2500). An oral informed consent was obtained from the participants prior to their enrolment in the project. Respondents first answered to a socio-demographic questions before sharing their opinion about various topics regarding their hearing dog. The interviews were video recorded, transcribed, and coded to produce the conclusions of this article.

### 2.3. Participants

Owners of hearing dogs are scarce in Quebec province. Thus, the number of recruited participants was set according to the maximum number of individuals living in Quebec, owning a hearing dog, and willing to participate we were able to find. Recruiting a higher number of participants would have been preferable, but was not realistically doable

considering only a few individuals own a hearing dog in Quebec. However, the sample is still considered adequate to inform the research question, as it consists of people, both men and women, who received a hearing dog as a hearing assistance aid.

Recruitment was supported by community organizations (Audition Québec, Lions Foundation of Canada) and audiologists working in a public rehabilitation center (CIUSSS Capitale-Nationale). Participants were all adults living in Quebec and had a sufficient knowledge of French to be able to participate to the interviews. Two of them owned a hearing dog at the time of the interview. In consideration to our sample size, we decided to include a participant who previously owned a hearing dog but returned it to the *Foundation*; he is now on the waiting list to obtain another one. All dogs were provided by the *Lions Foundation*. In addition, participants had beforehand experience with hearing aids (or cochlear implants) and EAS. A list of the participants' characteristics is presented in Table 1.

**Table 1.** Participants characteristics (*n* = 3).

| ID | Age (y) | Sex | Location | Occupation | Living Arrangement | Main Language Spoken | HL * Degree | HL Etiology | Duration of HL (y) | Hearing Aids (y Obtained) | Obtention of Hearing Dog |
|----|---------|-----|----------|------------|-------------------|---------------------|-------------|-------------|-------------------|--------------------------|--------------------------|
| P1 | 48 | Female | Saint-Lin, QC | Worker | Home (partner and child) | French | Moderate (R); Severe (L) | Sound trauma | 8 | Two hearing aids (2014) | 2016 |
| P2 | 73 | Male | Montreal, QC | Retired | Home (partner) | French | Profound (R/L) | Otological, neuroma, Meniere's | 9 | 1 cochlear implant (2015) | 2019 |
| P3 | 51 | Female | Blainville, QC | Business owner | Home (partner) | French | Moderate to severe (R/L) | Congenital, head trauma | >30 | Two hearing aids (1990) | 2019 |

* Hearing loss.

### 2.4. Data Processing and Analysis

An analysis procedure combining qualitative content analysis and interpretative phenomenological analysis was used [30–33]. Given the limited number of participants in our study, we made the decision to forego the use of a qualitative analysis software and instead focus on analyzing the data through more traditional methods. We used Microsoft Word to transcribe and code our data. Once the interviews were transcribed in a text document, pertinent statements to keep for further analysis were identified. An initial categorization system was established based on the topics covered in the interview grid, which were related to the components of the ICF. Data was manually labeled in the text document according to this categorization system. We improved the initial content categorization system with the new elements found during the data processing, such as the ones that came up spontaneously by the participants during the interviews (i.e., social acceptability, trusting the dog's abilities, etc.). All the statements were then transferred in a classification table according to these global categories, that became our units of content. Each of the kept statements was analyzed to identify the meaning attached. A code was attributed to each statement. For example, the statement "*Of course, to have to go to Ontario to get the dog, it was really easy. Over there, you have no problem with assistance dogs. None!*" (P1) was classified in the *Acquiring process (Ontario)* unit of content and was attributed the code *easiness*. Final units of content include Participant's profile (handicap situations, emotional perspective on hearing loss), Experience with hearing technologies (hearing aids/cochlear implant, assistance technology devices; EAS), Acquiring process (first heard of this service, expectations, procedures, Ontario), Experience with the hearing dog (tasks, non-auditory experience, negative elements, hearing technologies, responsibilities), Dog as a rehabilitation mean (social acceptability, recognition and visibility, trust in the abilities, satisfaction),

Quality of life (sense of security, social interactions, participation), and Acquisition system (role of the Foundation, considering a Quebec point of service). Codes vary depending on the unit of content the statements are found in. We examined the frequency of each code within each content unit to draw conclusions, taking into account any contextual factors that may be influencing the results, such as personal and environmental facilitators. The participants' experiences were analyzed to define common characteristics and connected to broader social, cultural, and political phenomena to produce the findings of this study [33]. Cross-verification was achieved through the research team transcribing, classifying, and coding the data. A reflexive approach was applied throughout the process to ensure quality and rigor.

## 3. Results

### 3.1. The Acquiring Process

The three participants interviewed learned about the existence of hearing dogs and the *Lions Foundation of Canada* in various ways, either by searching the Internet, by meeting with an owner of such dog or by recommendation of their attending physician. Their main expectations were replacing the EAS (P1), alerting when someone calls their name (P2) and alerting when the alarm clock rings (P3). All participants found that the pre-pairing process was quite long and complex, mainly because of the exhaustive documentation required. P2 judged that the communications with the organization were insufficient since he had no news throughout this wait. However, the pre-pairing home assessment was a step perceived positively by all, as was the mandatory training stay in Oakville, Ontario. The ease of the process, the good support and the free travel arrangements were the most salient elements mentioned. P2 and P3 said they had meaningful social connections with their peers while training, despite finding the learning experience physically and mentally demanding. P2 mentioned feeling nervous but found it reassuring that the training was accessible and adapted to people with hearing loss: "*[ . . . ] so if I lost something I had the visual support next to it*".

### 3.2. Experiencing the Hearing Dog

The main task performed by the participants' hearing dogs was alerting to repetitive sounds (their name, ringtone, emergency vehicle, door, washer/dryer, stove, microwave, alarm clock, fire alarm). P1 also mentioned the dog's ability to perceive alarms in background noise: "*If there is an alarm, [the dog] will perceive the alarm through the movie [at the cinema]*". P1 and P3 mentioned that their dog could warn them when their child cries or calls them from another room in the house. The dogs were also trained to respond to basic commands (sit, down, come, etc.) in French, English, and gestures. P1 also trained her dog for new tasks related to her workplace. All participants mentioned that their dog had been trained for tasks adapted to their needs and that there was nothing more the dog could do for them. More so, they claim that hearing dogs do a lot more than what they are trained to do. Nearly half of the statements about the experience of owning a hearing dog concerned non-auditory aspects. Among these, companionship is the most mentioned element, followed by the reduction of anxiety (moral support), improvement of physical health and motivation and reduction of loneliness. On this matter, it is impressive to note that a participant's dog saved her twice by warning her of a heart attack in progress.

The three participants mentioned an increase in their autonomy since acquiring their dog: "*It kind of freed my family. Completely also my entourage. This makes me feel less handicapped. You feel a lot more normal*" (P3). They also all reported a higher sense of security since acquiring the dog, particularly in relation to the detection of alarms. Having a hearing dog would also increase social interactions by promoting conversations with strangers or other dog owners. Thus, concerning participation, we obtained testimonials demonstrating either an augmentation in participation ("*Even if you have hearing aids, sometimes you miss bits of them. [ . . . ] That's where isolation appears. [ . . . ] Whereas with [dog's name], well, I don't have that isolation*", P1) or no effect on participation. Only P1 experienced serious participation

restrictions at her workplace because of her dog. On this subject, difficulties regarding access to buildings with the dog was a massive concern reported by all respondents: "*[It was] more than a huge difficulty in Quebec. [ . . . ] being refused to go somewhere because it is not a Mira dog [ . . . ]*" (P3). In terms of recognition and visibility, all expressed feelings of not being understood by Quebec government's instances, which seem, from the perspective of the participants, to recognize very little of the rights of people with hearing disabilities and even less of those who benefit from a hearing dog. Thus, the recognition, but also the lack of knowledge on the part of the Quebecers (audiologists, civilians, security guards, owners and employees of businesses) is a significant issue for the participants.

Owning a service dog sure comes with its own set of responsibilities. The most irritating commitment reported was the permanent and constant duty to take care of the dog. Otherwise, cleanliness and behavior (i.e., barking or being distracted outside the home) were also mentioned a lot. When asked about the negative aspects of owning such dog, respondents all mentioned having been bothered several times by strangers touching or talking to the dog while it was at work. They must therefore constantly explain the role of the dog and educate people on how to behave around an assistance dog. Access restriction to certain buildings was also discussed. Being authorized to have only one assistance dog at a time (even once the dog is retired), the limited space for the dog on the plane and the costs related to the animal have all been named once on the negative side of the experience. Despite these negative aspects, respondents reported having complete confidence in their dog's abilities and expressed overall satisfaction with the daily support provided by their dog: "*She is extraordinary. Extraordinary is not even the word. I don't even know how to explain it*" (P3). P1 and P3 also mentioned that those around them were satisfied with the arrival of the dog in the household. In the particular case of P2, pairing failure was a difficult event to go through: "*Well, I was definitely disappointed, you know. But I have a kind of small mourning, there you know, but after that I came back*". P2 had to return the dog due to cleanliness problems, but also because he found the dog was not performing well all the time. It was sleeping a lot and sometimes did not alert its owner when it was supposed to. Although he returned his initial dog, P2 appreciated the experience enough to request another one; he is currently on the *Foundation*'s waiting list.

The opinion that hearing aids or cochlear implants are essential tools to improve communication for a person with hearing loss is unanimous among the interviewees. Although the connectivity options available are appreciated, it turns out that these aids are not flawless: some sounds remain inaudible, and adaptation can be difficult. Hearing assistance technology such as EAS can be used as a complementary rehabilitation means to help alert the individual of inaudible sounds. P1 and P3 were unhappy with this system and stopped using it altogether because of its many inconveniences, even if it meant having to find other ways to get by or relying more on their support network. On the other hand, P2 reported understanding the usefulness of the EAS and continues to use it to this day, although he finds it to be a rather artificial technology. He sees a complementary utility in it, the dog being for him mainly a tool to counter social isolation.

### 3.3. The Lions of Canada Foundation's Role

Participants were asked to describe their experience in Oakville and their general satisfaction with the *Foundation*'s role. What was mentioned the most was the attention paid to pairing the animal with the right beneficiary and the quality of the post-pairing follow-up. The good coordination of services was appreciated by all. However, it should be mentioned that one of the participants experienced a difficult pairing and did not like how the situation was handled by the Foundation. Accessibility during training is another point which did not make consensus among participants. The Foundation's personnel are said to be trilingual (English, French, ASL), but P2 was disappointed with the poor quality of certain staff members' French. Additionally, sign language was used, but only in ASL, while the man would have been more comfortable with the Quebec sign language (LSQ). P1 and P3 had less profound hearing loss and did not have communication and accessibility

difficulties during training. Another negative point mentioned by all is the website's lack of user-friendliness.

To close this topic, all participants mentioned that the Foundation requires a high level of care for the animal (veterinarian visit every 6 months, weight monitoring, grooming, walking, work break, training/practice). Although, this is not perceived as a negative element for beneficiaries who want the good of their animal. To conclude the interview, all respondents expressed a keen interest in the installation of a Foundation training center in Quebec to ease access to the service and promote the recognition of hearing dogs in the province.

## 4. Discussion

This project focused on service dogs specializing in hearing loss and aimed to describe the experience of owners of this little-known means of hearing assistance. This research is founded on three semi-structured interviews conducted with adults who currently own or have owned a hearing dog. The small sample size was predictable since only a few individuals currently own a hearing dog in Quebec. There may be several reasons why there are few hearing dogs in this province, including the lack of a local provider, inadequate promotion and accessibility or the rare recommendation of this assistive help as a means of auditory rehabilitation. Maintenance costs associated with caring for this type of help could also be one of these reasons, as participants mentioned that food and veterinary services are expensive and that they do not receive the same monetary compensation from the government as owners of blind dogs. Nevertheless, the explorative aim of this research is satisfied by the extent of the sample size presented. We propose to discuss the benefits and disadvantages of owning a hearing dog, but also the participants' satisfaction, social participation, and sense of security since acquiring the dog. Interpreting the results in a multidimensional approach will allow us to observe the relevance of the hearing dog as rehabilitation means for adults with hearing loss.

The perceptions and attitudes reported by the respondents are consistent with the frequent consequences of hearing loss mentioned in the literature such as social isolation, anxiety, frustration, or the feeling of being a burden [1–3]. Participants also had several points in common with the individuals on the hearing dog waiting list interviewed by Smith et al. [9], including being regular users of hearing assistance technology and reporting some limitations in social and emotional functioning. However, our respondents' expectations regarding the dog's role and abilities seemed lower and more precise. Thus, the three participants easily named a single element which was for them the priority, while only one of the participants verbalized having an expectation of forming a team with the dog (companionship), an element that had been mentioned by all the participants in Smith et al. [9]. It appears that functional expectations were predominant in our respondents, compared to emotional expectations.

In regard to the results, the most salient benefits of owning a hearing dog can be summed up in four words: usefulness, safety, motivation, and companionship. These findings are consistent with much of the previous literature on the subject. Almost no effect on social interactions or social participation was reported by the individuals interviewed, except for the increase in autonomy to carry out certain activities more easily. This point stays a controversy element in literature as some have not found any effect of hearing dogs on social interactions [11,14].

The three interviewees demonstrated a high level of satisfaction. Their main expectations were met and exceeded. The dogs were trained in an exemplary and adapted manner, resulting in a strong trust from the owner and leading to an increased feeling of security both at home and during daily activities. On the other hand, various benefits reported were not related to the auditory sphere (i.e., complicity, control of anxiety, motivation, increased physical and mental health and breaking of loneliness), these results being consistent with the previous literature [13,19,20]. We can link these elements with the notion of companionship, a notion used in reference to the moral and emotional support that dogs

can provide [9]. Although hearing dogs are not specifically trained to reduce the beneficiary's anxiety, they still contribute to the reduction of anxiety related to disabling real-life situations. This turns out to be even more relevant since individuals with hearing loss are more likely to experience isolation and restricted participation than hearing people [1–3].

Social acceptance and recognition were the most discussed topics during the interviews. The participants repeatedly noted the lack of knowledge about assistance dogs among Quebecers. Furthermore, access to businesses and buildings appears to be the biggest issue for the owners. These situations seem particularly problematic in Quebec, compared to the other places visited by the individuals interviewed. According to them, the existence and visibility of Mira foundation's dogs could be part of the explanation since almost exclusively these dogs seem to be recognized by the general public. Constantly having to explain and justify the presence of the dog appears to weigh heavily on the daily life of the beneficiaries, even if the benefits of the dog remain greater. It should be remembered that in Quebec and Canada, no specific law makes it possible to circumscribe the use of hearing assistance dogs [25], which sometimes leaves room for discrimination: supplement charged to the hotel for cleaning, new employer requirements, access to housing, etc. It seems clear that the institutional recognition of hearing assistance dogs is lacking, especially when compared to that of guide dogs for the blind for which the beneficiaries receive, among other things, a tax deductible. Legislatively defined use of the hearing dog could improve citizen understanding of the services offered by these dogs.

We reiterate here that the main role of the hearing dog is in some ways comparable to that of an EAS (i.e., alerting at the presence of a sound signal). So why get a dog instead of a technological system? According to the participants, dogs have particularities the EAS has not. First, they can alert the beneficiary when someone calls him/her by her name. This point was mentioned many times by our respondents as a facilitator in social contexts, where they often miss many communication cues. Second, the dog is seen as less constraining for the rest of the household since it does not bark or disturb anyone when alerting its owner. Finally, and most importantly, the dog can follow the beneficiary in his/her daily activities outside of the house. This represents a huge advantage when taken from a social participation and inclusion viewpoint, but also from a security perspective. However, it is worth mentioning that recent technological devices, such as intelligent watches, are now capable of alerting the owner of certain signals outside of its house (ex: an incoming phone call). This could potentially provide an alternative means of being alerted for individuals with hearing loss.

Although the current acquisition process available to Quebecers requires doing business with an organization located in Ontario, we found that the service is nevertheless accessible from a financial and linguistic point of view for people from Quebec. It should be noted, however, that the participants were either retired, on sick leave or business owners, which eliminated or reduced the constraints related to employment (having to take a leave, loss of income, etc.). The individualized pairing process, the training offered to dogs and beneficiaries, as well as the post-pairing follow-up remain the strengths of the Foundation. It is a generally positive portrait that emerges from the process carried out to obtain a hearing assistance dog through the Lions Foundation of Canada.

Nevertheless, all participants were interested and in favor of the establishment of a system for acquiring hearing dogs in Quebec province. Two main reasons motivate this point of view. First, respondents believe that a Quebec service point would facilitate the acquisition process, but also the post-pairing follow-up. For instance, veterinary care could be provided for free or at a reduced rate and the staff would be more comfortable in Quebec French and LSQ. The other reason resides in the idea that a more local branch would allow the Quebec population to be more aware of the existence and role of hearing dogs. Participants did not feel that their dogs were recognized in the same way as dogs from other specialties or foundations. Having a distribution service located in Quebec would allow them to obtain better civic and organizational recognition, which would improve their inclusion in everyday society. This shows that local implementation entails more than

practical benefits for owners. In regard of their own context and particular needs, other jurisdictions could also benefit from local hearing dog providers for the previous reasons.

It is important to acknowledge that our study has certain limitations that should be considered when interpreting the results. We did not systematically control for variables such as personality or prior experience with hearing assistive technologies, which could have affected the participants' experiences. However, we did consider these factors when interpreting the data and recognized the potential impact they may have had on the results. We did not include a measure of QoL in our study, as it is typically assessed using a specific questionnaire or similar tool. While it would be interesting to examine thoroughly the impact of this intervention on QoL, it was beyond the explorative scope of our current study. Finally, it is also important to recognize that our sample size was relatively small, which may have limited the generalizability of our findings. In order to fully understand the potential value of this hearing assistance for the Quebec population, the main findings put forward by the three participants will have to be re-evaluated on a larger scale within the population to better demonstrate the relevance of this aid in this context. Despite these limitations, we believe that our study provides valuable insights.

## 5. Conclusions

The results of our study suggest that the acquisition process for hearing dogs was perceived positively due to the ease of the process, the availability of supportive resources, and the provision of free travel accommodations. These factors were identified as particularly beneficial to the participants. Once the pairing with the hearing dog was completed, participants mentioned their dog was trained to perform tasks tailored to their specific needs. Additionally, the participants seemed to feel that the hearing dogs were able to do a lot more for them than what was originally intended. Although the primary function of hearing dogs is to alert its owner to repetitive sounds, a significant portion of the collected data focused on non-auditory benefits. Among those, the most frequently mentioned was companionship, followed by reduced anxiety (moral support), improved physical health, increased motivation, and reduced feelings of loneliness. According to the three participants in our study, the dogs enabled them to have more independence and feel safer in their daily lives, but also facilitated conversations with strangers and other dog owners. These findings indicate that the impact of hearing dogs may extend beyond the auditory realm and imply elements that are directly related to well-being [34], suggesting hearing dogs can have a wide range of positive effects on the overall QoL of their owners.

Based on the interviews conducted, it appears that social understanding and lack of institutional recognition of hearing dogs were major concerns for the participants. Access to businesses and buildings was also identified as a particularly significant issue. This lack of recognition can create barriers for individuals with hearing loss who rely on these animals for support and highlights the need for more widespread awareness and acceptance of these dogs within institutions and the broader community. There is a need to increase awareness and understanding of the role and value of assistance dogs among the public. By doing so, we can help to ensure that individuals with hearing loss and their dogs have equal access to spaces and opportunities. One potential way to address this issue is through the establishment of a Foundation training center in Quebec, which could provide a resource for education and support for individuals with hearing loss and their hearing dogs.

Overall, the hearing dog is a relevant way to combine the benefits of technical assistance with the psychosocial and emotional support of a pet. Considering that rehabilitation options that take emotional and psychological needs into account are among the most effective to increase the user's QoL [11], hearing dogs should be considered as a treatment option more often by hearing health professionals. Although conventional hearing technologies, such as hearing aids, cochlear implants, and EAS remain essential for a better functioning on a daily basis for many patients, the acquisition of a hearing dog as a complement would also open the door to greater benefits, which may improve the user's commitment in his rehabilitation process. Our conclusions are relevant to the Quebec's

context but could be enlarged to other localities of the world in similar posture bodies. Future work should comprehensively detail the different steps in the process of assigning a hearing dog from the perspective of the provider. The analysis of the balance of costs and benefits at provincial and federal levels should also be explored.

**Author Contributions:** Conceptualization, A.L., M.-A.T. and M.H.; methodology, A.L., M.-A.T. and M.H.; formal analysis, A.L. and M.-A.T.; investigation, A.L. and M.-A.T.; resources, M.H.; data curation, A.L. and M.-A.T.; writing—original draft preparation, A.L. and M.-A.T.; writing—review and editing, A.L. and M.H.; visualization, A.L.; supervision, M.H.; project administration, M.H.; funding acquisition, M.H. All authors have read and agreed to the published version of the manuscript.

**Funding:** This research was funded through two grants held by Mathieu Hotton (Fonds de Recherche du Québec en Santé, grant #324155; Interdisciplinary Research Center on Rehabilitation and Social Integration and Centre Intégré Universitaire de Santé et de Services Sociaux de la Capitale-Nationale, grant #0101582).

**Institutional Review Board Statement:** The study was conducted in accordance with the Declaration of Helsinki and approved by the Institutional Review Board (or Ethics Committee) of CIUSSS Capitale-Nationale (protocol #2022-2500, date of approval: 17 March 2022).

**Informed Consent Statement:** Informed consent was obtained from all subjects involved in the study.

**Data Availability Statement:** Not applicable.

**Conflicts of Interest:** The authors declare no conflict of interest.

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
