# Peer review of "The Role and Relevance of Hearing Dogs from the Owner’s Perspective: An Explorative Study among Adults with Hearing Loss"

_audiolres, doi:10.3390/audiolres13010006_

Round 1

Reviewer 1 Report

I hope that the message of this article will reach a wide audience of audiologists, audiologists and people who help deaf people in various ways.
Dogs helping deaf people is something normal and obvious today. In contrast, the support of a dog for a deaf person is not a topic known to the general public.
I think that public awareness is important here, that is, publicizing the topic at least in the media. It is also important to fight for the owners of these dogs to have the same rights as blind people who are supported by their dogs.
Thank you for raising this topic.

Author Response

We would like to express our sincere gratitude to reviewer 1 for taking the time to read and comment our article. We hope that this article will contribute to raising awareness about this little-known topic. This is an important issue that we believe deserves more attention. We believe that by sharing our research and insights, we can help to increase understanding and support regarding hearing dogs. 

Reviewer 2 Report

1.     Were there other experiences in Canada? Has nothing been reported?

2.     How is the ICF model linked to a hearing dog? How was this approach used in the study?

3.     How was the data analyzed? for example, any software like Nvivo was used.

4.      The authors stated that information was coded; how was this done? "The Lions of Canada Foundation's role"? would be informative if the authors mentioned the final codes or were "The Acquiring process" and "Expiring the Hearing dog".

5.     Is there any information about the hearing device's previous experience? I mean, how was it, good experience, bad experience? This could have an impact on the Hearing dog experience. How important could be the user's personality?

6.     One of the participants return the dog; why?

7.     If coding was performed from the information collected, it would be beneficial to use them to report the results. Was this the case?  

8.      How did the authors estimate an increase in QoL? was this measured?

9.     Conclusions should stick to the results, for instance, L239: "alerting them to the repetitive sounds….." which is related to the main task referred by the owners; L250; "companionships is the most mentioned, reduction of anxiety…….". L270: "lack of knowledge on the part of the Quebecers……". The conclusion is blurred.

10.  The maintenance cost of the dog can be one reason/limitation for the low number of hearing dogs in Quebec.
